# Bleomycin for Percutaneous Sclerotherapy of Venous and Lymphatic Malformations: A Retrospective Study of Safety, Efficacy and Mid-Term Outcomes in 26 Patients

**DOI:** 10.3390/jcm10061302

**Published:** 2021-03-22

**Authors:** Franck Nevesny, Olivier Chevallier, Nicolas Falvo, Kévin Guillen, Alexandre Malakhia, Julie Pellegrinelli, Pierre-Olivier Comby, Bertille Bonniaud, Marco Midulla, Romaric Loffroy

**Affiliations:** 1Image-Guided Therapy Center, ImViA Laboratory-EA 7535, Department of Vascular and Interventional Radiology, François-Mitterrand University Hospital, 14 Rue Paul Gaffarel, BP 77908, 21079 Dijon, France; brothersfs@icloud.com (F.N.); olivier.chevallier@chu-dijon.fr (O.C.); nicolas.falvo@chu-dijon.fr (N.F.); kguillen@hotmail.fr (K.G.); alexander.malakhia@chu-dijon.fr (A.M.); julie.pellegrinelli@chu-dijon.fr (J.P.); marco.midulla@chu-dijon.fr (M.M.); 2Department of Neuroradiology and Emergency Radiology, François-Mitterrand University Hospital, 14 Rue Paul Gaffarel, BP 77908, 21079 Dijon, France; pierre-olivier.comby@chu-dijon.fr; 3Department of Dermatology and Genetics of Developmental Anomalies, UMR Inserm 1231, François-Mitterrand University Hospital, 14 Rue Paul Gaffarel, BP 77908, 21079 Dijon, France; bertille.bonniaud@chu-dijon.fr

**Keywords:** venous malformations, lymphatic malformations, percutaneous sclerotherapy, bleomycin, patient-reported outcomes

## Abstract

Percutaneous sclerotherapy is used to treat venous and lymphatic vascular malformations, which can cause significant discomfort and/or disfigurement. The purpose of this study is to describe the bleomycin sclerotherapy technique and to evaluate its clinical and radiological efficacy and safety. We retrospectively identified consecutive patients with venous malformations (VMs) and lymphatic malformations (LMs) who underwent bleomycin sclerotherapy in 2011–2020 at our institution. We collected the clinical and radiological success rates, complications and recurrences separately in the VM and LM groups. We identified 26 patients, 15 with VMs and 11 with LMs. The significant volume reductions obtained were 45% in the VM group and 76% in the LM group (*p* = 0.003 and *p* = 0.009, respectively). Significant reductions in discomfort/pain and in cosmetic disfigurement were obtained in both groups. An overall improvement was reported by 69% and 82% of patients in the VM and LM groups, respectively. No major complications occurred during the mean follow-up of 51 ± 34 months in the VM group and 29 ± 18 months in the LM group. A recurrence developed within 2 years in 23% of patients. Bleomycin is clinically and radiologically effective for the treatment of venous and lymphatic malformations, with a high level of patient safety.

## 1. Introduction

Venous and lymphatic malformations rare congenital defects resulting from abnormalities in vascular system development during the embryonic period. They are classified as low-flow vascular malformations [1]. They should not be confused with vascular tumours, such as haemangiomas, which are characterised by endothelial hyperplasia [2].

Venous malformations (VMs) are the most common, with an incidence of approximately 1 per 10,000 births, while lymphatic malformations (LMs) have an incidence of approximately 1 per 16,000 births [3]. These malformations enlarge with the growth of the child and may be diagnosed later on in life. VMs and LMs can cause discomfort and disfigurement, particularly when located near the surface of the skin and in the head and neck region. LMs are found on the proximal part of the neck in nearly 75% of cases [4,5]. Thus, the adverse effect on quality of life can be substantial.

No strong treatment recommendations have been issued to date regarding which therapy or sclerosing agent should be used as first-line treatment in clinical practice. Surgical excision is not the first-line treatment because it more often leads to complications (haemorrhage, scarring and complications due to general anaesthesia) than other treatment methods. In recent years, percutaneous sclerotherapy has shown many advantages over surgical treatment, making it the preferred technique for first-line treatment [6]. Many sclerosing agents are available on the market, but there is no consensus regarding which is best [7]. The choice of the sclerosing agent is made on a case-by-case basis depending on the size and type of lesion and on its location. Ethanol is the most commonly used sclerosing agent and produces excellent results, although it also has the highest complication rate [8]. Bleomycin is a sclerosing agent with one of the lowest complication rates. Adverse events have been both minimal and transient [9]. However, it may increase the risk of skin hyperpigmentation and ulceration if injected too superficially under the skin. Bleomycin is a cytostatic antineoplastic agent that acts on the mechanistic target of rapamycin (mTOR) pathway. It induces an endothelial-to-mesenchymal transition that enhances its scleroembolic effect [10]. The cumulative lifetime dose of bleomycin should not exceed 400 mg or 5 mg/kg to limit the risk of pulmonary fibrosis [11,12]. Because of this dose limitation, bleomycin should be used with caution to allow the treatment of VMs and LMs while minimising the total dose.

The objective of this study was to assess the clinical and radiological efficacy and safety of bleomycin sclerotherapy in 26 patients with VMs or LMs treated over a 9-year period.

## 2. Materials and Methods

### 2.1. Study Population

All bleomycin sclerotherapy procedures performed at our institution over the 9-year period from 2011 to 2020 were identified by searching the hospital’s electronic database. Institutional review board was not required for this study due to its retrospective nature, but our ethics committee approved the study. For this single-centre cohort study, informed patient consent was waived in compliance with French legislation on retrospective studies of anonymized data. Patients treated with other sclerosing agents for VMs or LMs were excluded.

### 2.2. Percutaneous Procedure

Before the procedure, blood cell and platelet counts were measured, and simple coagulation studies were performed. No further examination, especially of lung function, was performed given the extreme rarity of lung toxicity of bleomycin at low doses.

The procedure was performed under local anaesthesia in 17 (65%) patients and under general anaesthesia in 9 (45%) patients. Reasons for general anaesthesia were young age (6 patients younger than 8 years), poor patient cooperation and anticipated technical challenges due to location of the malformation at the head or neck. Access to the lesion was obtained using an 18- to 25-gauge needle, at one to five sites; ultrasound guidance was used in 12 (46%) patients. An iodinated contrast agent was then injected to observe the filling of the malformation under fluoroscopy. If there was evidence of extravasation, the access site was abandoned. The injectable solution was prepared by diluting 15 mg of bleomycin in about 20 mL of 9% NaCl. The sclerosing agent was injected then slowly and carefully into the malformation under fluoroscopic guidance without contrast agent. Indeed, the malformation was already enhanced by previous contrast injection and washed with the sclerosing agent alone. The mean procedure time was less than 15 min.

Each patient received an injection of 0.5 mg/kg of corticosteroids (Methylprednisolone^®^ Mylan; Mylan S.A.S, Saint-Priest, France) during the scleroembolisation procedure and was prescribed paracetamol-tramadol tablets (Ixprim^®^; Grünenthal, Aix-la-Chapelle, Germany) to be taken orally for 7 days after the procedure.

### 2.3. Data Collection and Follow-Up

For each patient, the following data were collected from the electronic medical record: age, sex, location of the lesion, symptoms, volume (mL) of the lesion before and after the end of the treatment, amount (mL) of bleomycin injected, number of bleomycin injections and cumulative bleomycin dose.

We collected the peri-procedural complications, i.e., complications occurring within 30 days after the procedure and divided them into major and minor complications according to the Society of Interventional Radiology (SIR) classification [13]. Minor complications required no treatment or resolved within 24 h with symptomatic drugs. Major complications required treatment for more than 24 h or were more severe or irreversible.

To collect patient-reported outcomes, each patient was contacted by telephone in October 2020 and asked to complete a simple, standardized questionnaire. Patients completed a questionnaire evaluating disease symptoms and treatment satisfaction. Furthermore, the long-term outcomes of Bleomycin sclerotherapy were investigated in terms of patient-perceived changes in health. Pain/discomfort before and after treatment was evaluated using a numerical rating visual analogue scale ranging from 0 (no pain) to 10 (extremely painful). Patients also rated the extent of perceived change in cosmetic appearance on a health insurance scale ranging from 1 (very mild) to 7 (very significant). Finally, overall patient satisfaction with the treatment results was assessed on a 5-level semantic scale, ranging from strongly improved to markedly worsened.

Treatment endpoint was defined as final volume reduction of more than 50% and/or clinical improvement of at least three clinical categories (pain/discomfort and cosmetic appearance) and/or two overall satisfaction categories. As long as the treatment was effective, it was continued until disappearance of the lesion. Recurrence was defined as any clinically significant increase in lesion volume.

A follow-up visit with magnetic resonance imaging (MRI) was scheduled 3 to 6 months after the treatment session.

### 2.4. Statistical Analysis

Descriptive statistics and parameters, such as frequencies and percentages, were used and provided in order to accurately describe our experience regarding the sclerotherapy procedure with bleomycin. Values are presented as number (%), mean ± SD or median (range).

In each of the two groups, the non-parametric Wilcoxon signed-rank test was used to compare the values at baseline and at last follow-up for lesion size (as measured objectively by MRI) and for pain/discomfort and change in cosmetic appearance (as assessed by the patient on the questionnaire). Differences between the two groups were compared the non-parametric Kruskal–Wallis test. Values of *P* below 0.05 were taken to indicate statistically significant differences.

## 3. Results

### 3.1. Patients and Malformations

We identified 26 patients treated by bleomycin sclerotherapy for VMs or LMs at our centre from 2011 to June 2020. During the same period, 162 patients were treated with other sclerosing agents for vascular malformations, mainly venous and small malformations. Among the 26 patients, 15 (57%) had VMs and 11 (43%) had LMs. Table 1 reports their main features. Mean follow-up was 51 ± 34 months (range: 12–110) in the VM group and 29 ± 18 months (range: 12–60) in the LM group.

Magnetic resonance imaging (MRI) was performed in 24 (92%) patients before the procedure to confirm the diagnosis. Each malformation was measured on a T2-weighted turbo spin-echo (TSE) or short TI inversion recovery (STIR) MRI sequence by volumetric analysis via a 3D multi-modality processing platform to ensure measurement reproducibility. Of the remaining two patients, one underwent ultrasonography and the other had no imaging study. The effect of treatment in this patient was then assessed visually.

Five (19%) patients, all in the VM group, had had sclerotherapy of the same lesion using polidocanol (Aetoxisclerol^®^2%; Kreussler Pharma, Paris, France) as the sclerosing agent. All these sclerotherapy procedures were performed more than 6 months before inclusion in our study.

Table 2 reports the mean lesion volume in both groups. Heterogeneity was considerable, with volumes ranging from 2 to 700 mL in the VM group and 7 to 4212 mL in the LM group. The volume differences explain the differences in the number of injections, with ranges from 1 to 6 in the VM group and 1 to 3 in the LM group. The decision to perform repeat bleomycin injections was based on the reduction in malformation volume and on the patient’s overall clinical improvement as defined above. When several injections were required, the procedures were scheduled 5 to 6 weeks apart to minimise the risk of side effects. The cumulative bleomycin dose for all patients was 450 mL in the VM group and 220 mL in the LM group, respectively.

Eight (30%) patients received additional treatment after the last bleomycin sclerotherapy procedure, five in the VM group and three in the LM group. Additional aetoxisclerol sclerotherapy sessions were provided to three VM-group patients and two LM-group patients. Ethanol was used in 1 patient in the VM group and surgery in 1 patient in the LM group. The volume of the different lesions secondary treated with other sclerosing agents was stable over time, with no significant reduction (less than 15%), except in the patient treated surgically.

### 3.2. Radiological Outcomes

A final MRI scan was obtained in 21 (81%) patients, 11 in the VM group and 10 in the LM group. Table 2 reports the volume reductions after the last bleomycin injection.

A significant reduction in lesion volume was observed in 17 patients (17/21, 82%) and the percentage reduction ranged from 2% to 97% in the VM group and 0% to 100% in the LM group (Figure 1 and Figure 2). In the VM group, 3/11 patients (27%) had a volume reduction greater than 70%, five (45%) had a reduction of 20% to 70% and three (27%) had less than 20% of reduction. In the LM group, 8/10 (80%) patients had a reduction greater than 70%, including three (3/8, 37%) with elimination of the malformation and two (25%) patients had less than 20% of reduction. None of the patients experienced any growth of the malformation after treatment. In the five remaining patients with no final MRI, two of the VM group and one of the LM group were markedly improved with a malformation which almost clinically disappeared. No data were available for the two remaining patients.

### 3.3. Patient-Reported Outcomes

The clinical outcomes were assessed by a telephone interview in 24 (92%) patients, 13 in the VM group and 11 in the LM group, in October, 2020. Each patient completed the standardised questionnaire described above. The two remaining patients had no telephone interview and no post-treatment MRI; they were considered as lost to follow-up. Table 2 reports the main findings.

On the pain/discomfort scale, eight (35%) patients (three in the VM group and five in the LM group) had a reduction ≥5 points compared to their baseline value, whereas three (13%) patients showed no improvement in pain/discomfort. An improvement in pain/discomfort versus baseline was obtained in 11 (74%) VM-group patients and nine (85%) LM-group patients

Cosmesis was significantly improved in both groups (Table 2). The improvement in cosmesis on the 1–7 scale was greater than three points in seven (63%) LM-group patients and 2 (16%) VM-group patients. There was no improvement in cosmesis in six (26%) patients, two in the VM group and four in the LM group.

Finally, of the 24 patients contacted, nine patients (37%) including six in the LM group (67%) and three in the VM group (23%) reported that they were “markedly improved” by the treatment compared to their baseline condition, nine patients (37%) reported being “slightly improved” (six in the VM group and 3 in the LM group) and six patients (25%) reported no overall change compared to their baseline condition.

Although no statistics were performed regarding the correlation between radiological and clinical outcomes, 44% of the nine patients in whom the volume reduction by MRI was less than 50% reported pain/discomfort relief of three points or more on the numerical rating scale.

A total of 18 patients reported an overall improvement (from slight to marked), nine patients (69%) in the VM group and nine patients (82%) in the LM group.

No patients reported an overall worsening from baseline.

### 3.4. Complications

All procedures were technically successful. As shown in Table 3, no major complications occurred during follow-up. The injections were performed as outpatient procedures, under local anaesthesia in 17 patients and under general anaesthesia in nine patients. Hospital stay length was 1.3 ± 0.6 days (range: 1–3) in the VM group and 1 ± 0 day (range: 1–1) in the LM group. In the VM group, one patient had a 3-day hospital stay due to swelling in the ear, nose and throat that required prolonged clinical monitoring.

Minor transient post-procedural complications occurred in eight (30%) patients, four in each group. Among them, the most common were swelling, skin hyperpigmentation and pain (8%). Infection of the puncture site occurred in one patient in the LM group and resolved with antibiotic therapy.

In all, six (23%) patients experienced recurrences. The recurrences were noted at six months in one LM-group patient and at 2 years in three VM-group and two LM-group patients.

### 3.5. Comparison of the Groups with Venous Versus Lymphatic Malformations

The treatment responses were significantly better in the LM group than in the VM group, notably regarding the reduction in pain/discomfort (5.73 points versus 2.85 points, *p* = 0.019) and the improvement in cosmesis (3.28 points versus 1.08 points, *p* = 0.016). The reduction in lesion volume was also greater in the LM group than in the VM group (76% versus 45%), although the difference was not significant (*p* = 0.34), perhaps due to the small sample size. The LM group required less bleomycin than did the VM group (mean dose per patient, 20 ± 10 mL (range: 10–45) versus 30 ± 22 mL (range: 15–60)) and also had fewer injections (1.3 versus 2.1). No statistical test was performed for these parameters.

## 4. Discussion

In our study, one or more bleomycin injections proved effective in decreasing the volume of VMs and LMs. The volume decrease was greater for LMs than for VMs. Given the considerable variability in lesion size at baseline, the effect of the treatment was best expressed as the percentage of size reduction. The reduction was at least 50% for 64% of the VMs and 87% of the LMs. There were no major complications and minor complications were rare. Significant improvements were obtained regarding pain/discomfort, cosmesis and overall improvement in the patient-perceived changes in health and treatment satisfaction.

Venous and lymphatic malformations are rare congenital conditions whose causes and risk factors are largely unknown. They are generally present from birth and enlarge with age. Klippel–Trenaunay syndrome (KTS) is a rare congenital disease (1/30,000 births) characterized by a triad of cutaneous capillary malformations, varicose veins and hypertrophy of the bone and soft tissues [12]. In our cohort, 1 patient in the VM group had KTS.

Lesion volume is generally used to assess the efficacy of treatments for VMs and LMs. In most studies, a reduction of at least 50% defined a good to excellent result and we used the same threshold. As shown in Table 4, our results were similar to those in the literature, with a greater than 50% volume reduction of about 82% of LMs versus 83% on average for 11 studies included in a meta-analysis [14]. Regarding the VM group, our results were worse than those in the literature, with a greater than 50% reduction in only 64% of patients compared to 84% on average in 10 published studies [14]. This result may be partially explained by the absence of MRI follow-up data for 2 VM-group patients who saw significant clinical improvements following the treatment and who would have had a good to excellent MRI result. Two more patients had no MRI at all.

However, it should be noted that lesion volume on MRI does not always correlate with symptomatic improvement [15], although most studies used volume reduction as the primary endpoint. We therefore included a questionnaire to assess patient-reported outcomes including pain/discomfort, cosmetic disfigurement and overall improvement in quality of life. Of note, four (44%) of the nine patients in whom the volume reduction by MRI was less than 50% reported pain/discomfort relief of three points or more on the numerical rating scale. Only scant data are available on cosmesis or symptom-related quality of life, but a prospective study in 95 paediatric patients treated with bleomycin injections for haemangiomas and vascular malformations showed a 67% improvement in cosmesis [11], which is similar to our result.

Patients tended to perceive subtle changes rather than drastic improvements in their overall condition. Thus, only 37% of our patients reported a marked improvement in overall satisfaction. In contrast, in a study where the physicians evaluated the improvement, they found an excellent overall response for 87% of VMs and 84% of LMs [16]. This discrepancy underlines the importance of collecting patient-reported outcomes. Most of our patients had expectations that exceeded what can be realistically achieved. Complete long-term remissions are rare, recurrences are common and occur within a short time frame and several interventions are generally necessary over the patient’s lifetime to alleviate the symptoms and improve cosmesis. The lifelong nature of venous and lymphatic malformations should be explained to the patients at the time of the diagnosis and feasible improvements should be described so that patients can adjust their expectations to reality.

None of our 26 patients experienced major complications and eight (30%) had minor complications that resolved fully. A retrospective study of 55 patients with VMs reported comparable results, with minor complications in 10 (18.2%) patients [17]. A meta-analysis of 27 studies (1325 patients) treated with bleomycin reported a total of 192 adverse events (14%). The most common complications were swelling (10%), fever (4%) and wound infection (1%). Transient hyperpigmentation occurred in 0.8% of cases [14].

Bleomycin was first used as an anti-tumour agent in the 1960s [16]. It was first used as a sclerosing agent in 1977 to treat cystic lymphangioma [18]. Bleomycin is associated with a risk of pulmonary fibrosis when given in large cumulative doses (> 400 mg during the lifetime or a single 30 mg dose). The risk is greatest in elderly patients, patients with kidney disease and patients with cancer undergoing systemic bleomycin therapy [19,20]. The mean cumulative intra-lesional dose in our study was 30 ± 22 mL per patient in the VM group and 20 ± 10 mL in the LM group, i.e., far lower than the doses associated with pulmonary fibrosis. We are aware of a single report of acute pulmonary toxicity after bleomycin sclerotherapy, in an 8-month-old infant who received an intra-lesional total dose of 7 mg and recovered fully without developing pulmonary fibrosis [21]. No cases of lung toxicity occurred in our patients. Nevertheless, a larger cohort study with a longer follow-up may be necessary to assess the prevalence of this complication.

The efficacy of bleomycin also compares well with that of other commonly used sclerosing agents. Ethanol has long been recognized as the strongest sclerosing agent, with a remission rate ranging from 75% to 95% [11] and is widely used due to its availability, low cost and broad indications [22]. However, since 2016, several studies have shown that bleomycin is non-inferior to ethanol. In a meta-analysis, bleomycin provided similar volume reductions as did other sclerosing agents but with fewer complications (odds ratio, 0.1; 95% CI, 0.03–0.39) [14].

In addition, a recent meta-analysis of 32 studies included 1121 patients with veno-lymphatic malformations of the head and neck, treated with different sclerosing agents (bleomycin, pingyangmycin, ethanol and sodium tetradecyl sulphate sclerotherapy) [23]. Of the 1121 patients, bleomycin was used for 562 patients and pingyangmycin for 559 patients. A lesion reduction was found in 93.7% of patients with a mean number of sclerotherapy sessions per patient of 3.4 (range: 1–9). Minor complications occurred in 16.2% of patients while major complications occurred in only four (1.1%) patients. Pingyangmycin sclerotherapy achieved subjective or objective lesion size reduction in 96.3% [23]. Another meta-analysis of nine retrospective cohort studies involving 132 patients suffering from orbital and peri-orbital veno-lymphatic malformations assessed results of sclerotherapy [24]. When LMs and VMs were treated with bleomycin, objective reduction was achieved for 78.5% and 100% of the patients, respectively. When LMs and VMs were treated with pingyangmycin, objective reduction was achieved for 96% et 100%, respectively. Minor complications rate was less than 10% for pingyangmycin and around 26% with bleomycin. No major complications were reported with the use of pingyangmycin [24].

In practice, bleomycin sclerotherapy as the first choice in our study was at the discretion of the interventional radiologist. However, our main criteria for choosing between bleomycin and polidocanol as first line treatment was based on the lesion volume. In case of small lesions, polidocanol was preferred because of the risk of being not able to inject a sufficient volume of bleomycin. At the opposite, bleomycin was preferred in case of larger lesions because of the impossibility of injecting safely a large amount of polidocanol. Then, bleomycin was not used as first-line, but mainly dedicated to vascular malformations which were potentially contraindicated to polidocanol sclerotherapy.

Our study has the limitations inherent in the retrospective design. First, the sample size was small, reflecting the rarity of vascular malformations. Second, no subclassification of the type of VMs and LMs was performed given the small number of patients, preventing outcome assessment by malformation subtype. Third, the variability in lesion size made volumetric data analysis heterogeneous. To overcome this limitation, the effect of the treatment was best expressed by the median volume reduction in percentage. Fourth, five patients did not undergo a final MRI assessment. Fifth, the patient-reported outcomes were collected in October 2020, several months to several years after the end of the treatment, creating a risk of recall/memory bias. However, the vast majority of sclerotherapy procedures took place after 2015 and were performed in young patients (average age < 30 years old). These patients were very involved in the course of their disease, having accurate knowledge of their symptoms and treatments. Furthermore, two patients did not complete the questionnaire. Finally, the questionnaire did not list all the symptoms experienced by the patients.

## 5. Conclusions

Although our study included a limited number of patients, it suggests that percutaneous bleomycin is safe and effective for the treatment of VMs and LMs. The lesion size reduction was 50% or more for 64% of the VMs and 87% of the LMs. Most patients reported improvements in pain/discomfort and cosmesis, although only a minority felt markedly improved. The patient-reported outcomes did not consistently reflect the efficacy in terms of lesion size reduction. Finally, further research is needed to understand why bleomycin sclerotherapy was more effective in the LM group than in the VM group.

## Figures and Tables

**Figure 1 jcm-10-01302-f001:**
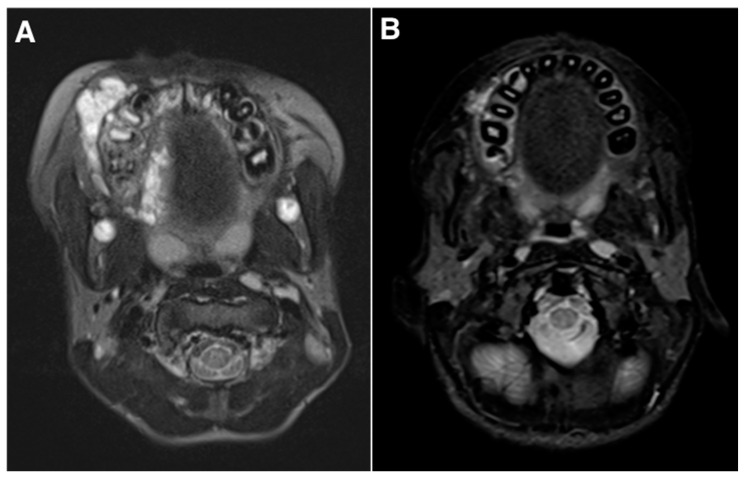
Images from a 32-year-old male with symptomatic venous malformation (VM). (**A**) pre-treatment T2-weighted STIR magnetic resonance imaging (MRI) showing an extensive VM of the right cheek cantered on the upper jaw. (**B**) Post-treatment T2-weighted MRI after two sessions of bleomycin sclerotherapy showing significant volume reduction of the VM (about 70%), with significant discomfort (visual analog scale, 2 versus 5), cosmesis score (2 versus 6) and overall satisfaction (markedly) improvement.

**Figure 2 jcm-10-01302-f002:**
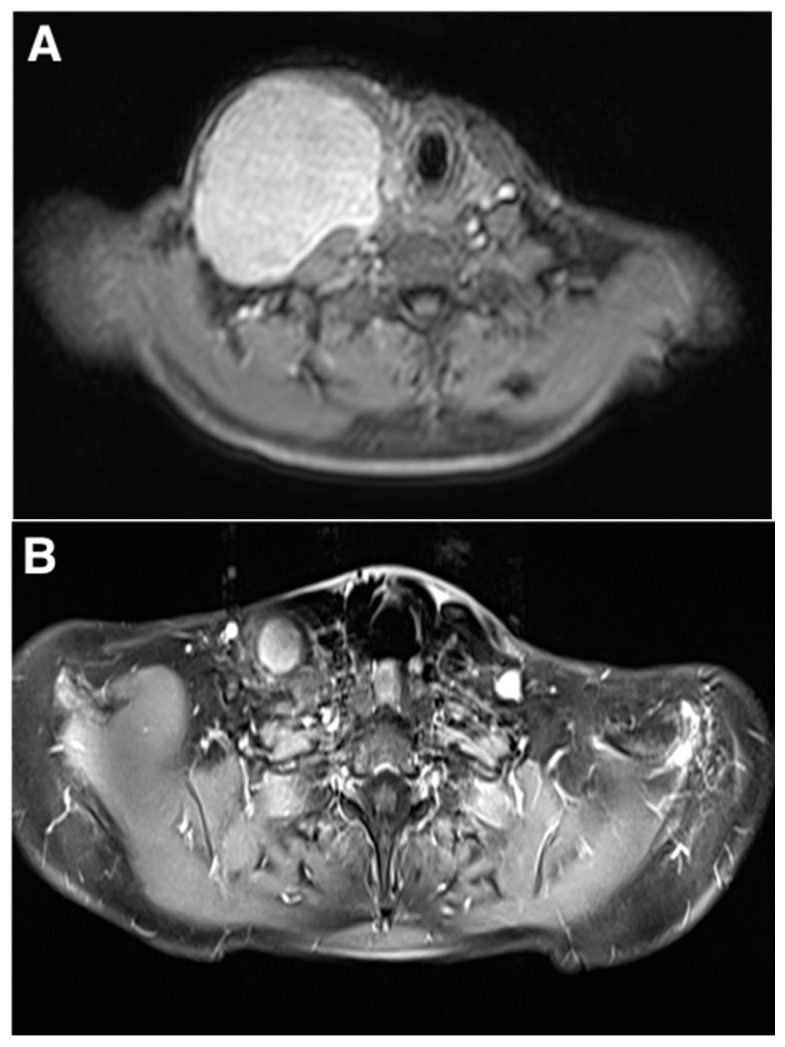
Images from a 39-year-old female with symptomatic lymphatic malformation (LM). (**A**) Pre-treatment T1-weighted post-gadolinium magnetic resonance imaging (MRI) showing an extensive cystic lymphangioma of the right supraclavicular fossa and neck. (**B**) Post-treatment T1-weighted MRI post-gadolinium after two sessions of bleomycin sclerotherapy showing significant volume reduction of the LM (about 80%) with significant discomfort (VAS, 1 versus 6), cosmesis score (1 versus 7) and overall satisfaction (markedly) improvement.

**Table 1 jcm-10-01302-t001:** Characteristics of the 26 patients and sclerotherapy procedures.

Clinical Data	VM	LM
n	15	11
Age (years)	30.8 ± 18.6 (8–59)	23.3 ± 16.2 (1–43)
Sex		
Male	6 (40)	6 (54.5)
Female	9 (60)	5 (45.5)
Location		
Head and neck	10 (66.7)	5 (45.5)
Extremity	4 (26.7)	2 (18.2)
Trunk	1 (6.6)	4 (26.7)
Previous treatment		
Aetoxisclerol sclerotherapy	5 (33.3)	0 (0)
Surgery	0 (0)	0 (0)
Laser	0 (0)	0 (0)
Additional treatment		
Aetoxisclerol sclerotherapy	3 (20)	2 (18.2)
Ethanol	2 (13.3)	0 (0)
Surgery	0 (0)	1 (9.1)
Primary symptom		
Pain	8 (53.3)	4 (26.7)
Swelling	15 (100)	11 (100)
Bleeding	1 (6.7)	1 (9.1)
Treatment sessions	2.1 ± 1.5 (1–6)	1.3 ± 0.7 (1–3)
Sclerotherapy volume per patient (mL)	30.0 ± 22.0 (15–60)	20.0 ± 10.5 (10–45)

VM, venous malformation; LM, lymphatic malformation; n, number. Values are as n (%) or mean ± SD (range).

**Table 2 jcm-10-01302-t002:** Radiological results, clinical outcomes and overall satisfaction.

Variable	VM	LM	*p* Value
Lesion MRI volume (mL)			
n	11	10	
Pre-treatment volume	37/142 ± 242 (3–645)	41/545 ± 1379 (7–4212)	
Posttreatment volume	21/136 ± 216 (0.2–480)	7/532 ± 1487 (0–4212)	
Volume reduction	45 ± 27.8 (2–97)	83 ± 39 (0–100)	0.340
*p* value	0.003	0.009	
Volume reduction > 50%	7 (64)	8 (80)	
Pain/discomfort			
n	13	11	
Pre-treatment VAS	6.0 ± 1.6 (4–8)	7.6 ± 2.5 (0–10)	
Posttreatment VAS	3.2 ± 2.1 (0–7)	1.9 ± 2.1 (0–6)	
VAS reduction	2.8 ± 2.1 (0–6)	5.7 ± 3.1 (0–10)	0.019
*p* value	0.002	0.004	
Cosmesis			
n	13	11	
Pre-treatment score	4.5 ± 2.3 (1–7)	5.0 ± 1.9 (2–7)	
Posttreatment score	3.4 ± 2.2 (1–6)	1.7 ± 1.0 (1–3)	
Score improvement	1.1 ± 1.5 (0–5)	3.3 ± 2.3 (0–6)	0.017
*p* value	0.016	0.005	
Overall satisfaction			
n	13	11	
Markedly improved	3 (23.0)	6 (54.5)	
Slightly improved	6 (46.2)	3 (27.3)	
Markedly + slightly improved	9 (69.2)	9 (81.8)	0.116
No change	4 (30.8)	2 (18.2)	
Slightly worsened	0 (0)	0 (0)	
Markedly worsened	0 (0)	0 (0)	

VM, venous malformation; LM, lymphatic malformation; MRI, magnetic resonance imaging; mL, millilitres; VAS, visual analog scale; n, number. Values are as n (%) or median/mean ± SD (range). Values of *p* below 0.05 were taken to indicate statistically significant differences.

**Table 3 jcm-10-01302-t003:** Complications and follow-up.

Variable	VM	LM
n	15	11
Major complications	0 (0)	0 (0)
Minor complications	4 (26.7)	4 (36.4)
Pain	1 (6.7)	1 (9.1)
Skin pigmentation	1 (6.7)	1 (9.1)
Swelling	2 (13.3)	1 (9.1)
Bleeding	0 (0)	0 (0)
Nausea	0 (0)	0 (0)
Fever	0 (0)	0 (0)
Local infection	0 (0)	1 (9.1)
Hospital stay (days)	1.3 ± 0.6 (1–3)	1.0 ± 0.0 (1–1)
Follow-up (months)	51 ± 34 (12–120)	29 ± 18 (12–60)
Recurrence		
At 6 months	0 (0)	1 (9.1)
At 24 months	3 (20)	2 (18.2)
At mean follow-up	3 (20)	2 (18.2)

VM, venous malformation; LM, lymphatic malformation; n, number. Values are as n (%) or mean ± SD (range).

**Table 4 jcm-10-01302-t004:** Outcomes of bleomycin sclerotherapy for VM and LM in main series from the literature.

Author, Year	Definition of Good to Excellent Improvement	N Total	N According to Definition	% According to Definition
**VM**				
Zhi et al., 2007 [25]	>50%	82	79	96
Chen et al., 2010 [26]	>50%	11	10	91
Sainsbury et al., 2011 [27]	>60%	42	38	90
Spence et al., 2011 [28]	>50%	17	15	88
Yue et al., 2013 [29]	>50%	12	11	92
Zhang et al., 2013 [10]	>80%	63	41	65
Bai et al., 2014 [30]	>60%	40	39	97
Jia et al., 2014 [31]	>50%	33	31	94
Mohan et al., 2014 [32]	>50%	27	24	89
Ul Haq et al., 2015 [33]	>50%	14	8	57
Songsaeng et al., 2015 [34]	>75%	16	12	75
Shigematsu et al., 2018 [35]	>50%	18	18	100
Sindel et al., 2018 [36]	>50%	20	17	85
Ahmad et al., 2019 [37]	>75%	35	16	45
Helal et al., 2019 [38]	>60%	30	30	100
Mean	-	31	26	84
Our study	>50%	11	7	64
**LM**				
Orford et al., 1995 [39]	>50%	16	14	88
Zulfiqar et al., 1999 [40]	>50%	11	8	73
Kim et al., 2004 [41]	>50%	10	6	60
Mathur et al., 2005 [42]	>50%	7	4	57
Rawat et al., 2006 [43]	>50%	19	16	84
Niramis et al., 2010 [44]	>50%	70	58	83
Rozman et al., 2011 [45]	>50%	24	20	83
Sandlas et al., 2011 [46]	>50%	10	9	90
Kumar et al., 2012 [47]	>50%	35	33	94
Erikci et al., 2013 [5]	>50%	14	12	86
Chaudry et al., 2014 [48]	>90%	31	12	38
Raichura et al., 2017 [49]	>60%	13	12	92
Porwal et al., 2018 [50]	>50%	8	8	100
Nuruddin et al., 2019 [51]	>70%	12	12	100
Mean	-	21	18	82
Our study	>50%	10	8	80

VM, venous malformation; LM, lymphatic malformation; N, number; %, percentage.

## Data Availability

The data presented in this study are available on request from the corresponding author. The data are not publicly available due to identity reasons.

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
