# Peer review of "Bleomycin for Percutaneous Sclerotherapy of Venous and Lymphatic Malformations: A Retrospective Study of Safety, Efficacy and Mid-Term Outcomes in 26 Patients"

_jcm, 2021, doi:10.3390/jcm10061302_

Round 1
Reviewer 1 Report
Kindly comment on the following
- 3rd paragraph in Introduction - 'No strong treatment recommendation.........' what do you exactly mean by that?
- Percutaneous procedure 2.2. - .......injectable solution prepared by diluting ..................... under fluoroscopic guidance - You have not mentioned if you have mixed any contrast with the solution to be visualised under fluoroscopy.
- Did you subclassify the VMs into type 1 -type 4 or into subcutaneous, intramuscular, intraosseous and LMs into macro- and micro-cystic and if that has any effect on the outcome.
- There are limitations on volumetric data analysis of vascular malformations follow up, particularly of the diffuse varieties. How did you overcome such limitations.
- Eight (8) patients (30%) received additional treatment after last Bleomycin therapy. Did it have any effect on the outcome?
- Criteria for completion of treatment and recurrence are not described. Kindly describe treatment end point and recurrence.
- Last paragraph of 3.3. A total of 16 pt. or 18 patients?
- Although you have mentioned that radiological and clinical outcomes do not always correlate, no statistics have been provided for the same.
- In Discussion, size and volume have been used interchangeably. Kindly clarify.
- Quality of life is not defined in the study and therefore significant improvement in QoL should not be mentioned unless specified further.
- In Results section you have mentioned 'Mean cumulative dose for VM of 450ml... and in Discussion you have written mean cumulative dose to be 25 mg. Kindly specify in each relevant section.
Author Response
Kindly comment on the following:
1. 3rd paragraph in Introduction - 'No strong treatment recommendation.........' what do you exactly mean by that?
Response: Thank you very much for your comments. We mean that there is no precise recommendation regarding which treatment should be used as first-line treatment in clinical practice. The decision is made on a case-by-case basis according to the risk-benefit ratio of each sclerosing agent used in interventional radiology. It has been clarified in the text.
2. Percutaneous procedure 2.2. - .......injectable solution prepared by diluting ..................... under fluoroscopic guidance - You have not mentioned if you have mixed any contrast with the solution to be visualised under fluoroscopy.
Response: Thank you very much for your comments. No, we did not use contrast agent during injection of sclerosant but just before to fill in the malformation. The sclerosing agent was then injected slowly and carefully into the malformation under fluoroscopic guidance without contrast agent. Indeed, the malformation was already enhanced by previous contrast injection and washed out with the sclerosing agent alone. It has been clarified in the text in the 2.2 section, 2ndparagraph.
3. Did you subclassify the VMs into type 1 -type 4 or into subcutaneous, intramuscular, intraosseous and LMs into macro- and micro-cystic and if that has any effect on the outcome.
Response: Thank you very much for your comments. No, we did not use a subclassification given the small number of patients and the potential lack of power in terms of statistics. We agree that it could have impact on the outcomes. This limitation has been added in the limitations section of the manuscript.
4. There are limitations on volumetric data analysis of vascular malformations follow up, particularly of the diffuse varieties. How did you overcome such limitations.
Response: Thank you very much for your comments. We fully agree with this comment. It was due to the variability in lesion size. Indeed, in both groups (VMs and LMs), some lesions measured for example up to 4000cc and others only 20cc. To overcome this limitation, the effect of the treatment was best expressed by the median volume reduction in percentage instead of mean volume reduction. This is clarified in the legend of table 2 and added in the limitations section of the manuscript.
5. Eight (8) patients (30%) received additional treatment after last Bleomycin therapy. Did it have any effect on the outcome?
Response:Thank you very much for your comments. Exactly, 8 patients received additional treatment after the last Bleomycin injection. We did not identify any effect on the outcome. The volume of the lesions was stable, with no significant reduction with other agent (less than 15%) except for the only patient who received surgical treatment. It has been clarified at the end of the 3.1 paragraph.
6. Criteria for completion of treatment and recurrence are not described. Kindly describe treatment end point and recurrence.
Response: Thank you very much for your comments. You’re right. Completion of treatment or end-point and recurrence have been described in the 2.3 section in the 4thparagraph. Treatmentendpoint was defined as final volume reduction of more than 50% and/or clinical improvement of at least three clinical categories (pain/discomfort and cosmetic appearance) and/or two overall satisfaction categories. As long as the treatment was effective, it was continued until the lesion had completely disappeared. Recurrence was defined as any clinically significant increase in volume.
7. Last paragraph of 3.3. A total of 16 pt. or 18 patients?
Response: Thank you very much for your comments. Sorry about that. Indeed, this is 18. It has been corrected.
8. Although you have mentioned that radiological and clinical outcomes do not always correlate, no statistics have been provided for the same.
Response: Thank you very much for your comments. We fully agree. No statistics have been performed on this subject. However, 44% of the 9 patients in whom the volume reduction by MRI was less than 50% reported pain/discomfort relief of 3 points or more on the numerical rating scale. It has been added in the 3.3 section.
9. In Discussion, size and volume have been used interchangeably. Kindly clarify.
Response: Thank you very much for your comments. It has been modified accordingly. Volume has been used through the all discussion section instead of size.
10. Quality of life is not defined in the study and therefore significant improvement in QoL should not be mentioned unless specified further.
Response: Thank you very much for your comments. We fully agree. QoL has not been directly assessed. Only patient-perceived changes in health and overall satisfaction have been evaluated. It has been clarified in the 2.3 section, paragraph 3. It has been modified at the end of the 1stparagraph of the discussion section too.
11. In Results section you have mentioned 'Mean cumulative dose for VM of 450ml... and in Discussion you have written mean cumulative dose to be 25 mg. Kindly specify in each relevant section.
Response: Thank you very much for your comments. We fully agree. Indeed, 450 mL equals to the cumulative dose in the VM group (all patients) and 220 mL in the LM group (all patients). It has been clarified at the end of the 3.1 paragraph. On the other hand, 25 mL corresponds to the average cumulative dose per patient in the two groups. This number has been removed. Indeed, we modified this for each group in the discussion section (page 10) to make it easier to understand. The mean cumulative intra-lesional dose in our study was 30±22 mL (range, 15-60) per patient in the VM group and 20±10 mL (range, 10-45) in the LM group. It has been added in the 3.5 paragraph too.
Furthermore, the manuscript has been double checked by a native speaker for English language.
Reviewer 2 Report
I read the original article by Nevesny and colleagues on their experience with bleomycin for percutaneous sclerotherapy of VM and LM with interest. VM and LM are relatively uncommon conditions, often with uncertain treatment outcomes. The literature on this is scarce too, hence clinical studies assessing treatment outcome on VM and LM are often helpful.
I have several comments and questions for the authors, hopefully they will help to improve the article further.
- How did the authors decide on which patients with VM/LM should be treated with bleomycin (and other sclerosants) in this study? How many patients with VM/LM had sclerotherapy (all sclerosants) during this period? Was bleomycin the authors’ first line sclerosant for VM and LM?
- What were the pre-operative and post-operative investigations that the authors perform before and after bleomycin sclerotherapy; for e.g. any screening of lung function?
- I presume the collection of patient reported outcome by telephone in October 2020 was, at least partially carried out for the purpose of this study. Therefore, this study was not entirely retrospective, and I believe a local research ethics consideration should be carried out, i.e. not waived. Please comment.
- The collection of patient reported outcomes by telephone in October 2020 would carry significant memory biases particularly for those patients who had the procedure several years ago (since 2011!). For e.g. the accuracy based on the remembering the VAS for pain pre- and post-treatment for something, let’s say 9 years ago would be questionable. Furthermore, some of these patients had multiple sclerotherapy sessions with different sclerosants, hence it would be difficult to ascertain that the outcomes were directly resulted from bleomycin. Please comment and these biases should be included in the discussion as limitations.
- Based on the findings of this study, would the authors advise more bleomycin to be considered for VM and LM? Any specific advice for bleomycin treatment for VM and LM that this study added which the readers could learn from; particularly pulmonary fibrosis which would be the most feared complication for bleomycin?
Author Response
I read the original article by Nevesny and colleagues on their experience with bleomycin for percutaneous sclerotherapy of VM and LM with interest. VM and LM are relatively uncommon conditions, often with uncertain treatment outcomes. The literature on this is scarce too, hence clinical studies assessing treatment outcome on VM and LM are often helpful.
I have several comments and questions for the authors, hopefully they will help to improve the article further.
1. How did the authors decide on which patients with VM/LM should be treated with bleomycin (and other sclerosants) in this study? How many patients with VM/LM had sclerotherapy (all sclerosants) during this period? Was bleomycin the authors’ first line sclerosant for VM and LM?
Response:Thank you very much for your comments. Bleomycin sclerotherapy as the first choice was at the discretion of the interventional radiologist. However, our main criteria for choosing between bleomycin and aetoxisclerol as first line treatment is based on the lesion volume. In case of small lesions, aetoxisclerol was preferred because of the risk of being not able to inject a sufficient volume of bleomycin. At the opposite, bleomycin was preferred in case of larger lesions because of the impossibility of injecting safely a large amount of aetoxisclerol. Overall, during the study period, 216 patients were treated with other sclerosing agents for vascular malformations, mainly venous and small malformations. Then, bleomycin was not used as first-line, but mainly dedicated to vascular malformations which were contraindicated to aetoxisclerol sclerotherapy. All these statements have been added in the 2.1 materials and methods section, in the 3.1 result section, and in the last part of the discussion section of the manuscript.
2. What were the pre-operative and post-operative investigations that the authors perform before and after bleomycin sclerotherapy; for e.g. any screening of lung function?
Response: Thank you very much for your comments. Before the procedure, blood cell and platelet counts were measured and simple coagulation studies were performed.No further examination of lung function due to the extreme rarity of lung toxicity of bleomycin at low doses. It has been added in the text in the 2.2 section.
3. I presume the collection of patients reported outcome by telephone in October 2020 was, at least partially carried out for the purpose of this study. Therefore, this study was not entirely retrospective, and I believe a local research ethics consideration should be carried out, i.e. not waived. Please comment.
Response: Thank you very much for your comments. This study was retrospective, even for collection of data on the long-term as part of routine clinical practice. According to our legislation, no IRB is necessary in such a setting and our institutional ethics committee systematically approve this kind of retrospective study. It has been clarified in the IRB statement at the end of the manuscript and in the 2.1 paragraph, as suggested.
4. The collection of patients reported outcomes by telephone in October 2020 would carry significant memory biases particularly for those patients who had the procedure several years ago (since 2011!). For e.g. the accuracy based on the remembering the VAS for pain pre- and post-treatment for something, let’s say 9 years ago would be questionable. Furthermore, some of these patients had multiple sclerotherapy sessions with different sclerosants, hence it would be difficult to ascertain that the outcomes were directly resulted from bleomycin. Please comment and these biases should be included in the discussion as limitations.
Response: Thank you very much for your comments. We fully agree. However, the vast majority of sclerotherapy procedures took place after 2015 and in young patients (average age less than 30 years old). Those patients were symptomatic and very involved in the course of their treatment, having accurate knowledge of their disease, treatment and symptoms. Of course, it can be a bias and it has been added in the limitations section, as suggested.
5. Based on the findings of this study, would the authors advise more bleomycin to be considered for VM and LM? Any specific advice for bleomycin treatment for VM and LM that this study added which the readers could learn from; particularly pulmonary fibrosis which would be the most feared complication for bleomycin?
Response: Thank you very much for your comments. We did not find any serious complications in this study, and with good results. The benefit-risk ratio of bleomycin is favorable. Pulmonary toxicity is clearly not a complication to be feared with such low doses. Only one case of pulmonary toxicity induced by bleomycin has been reported in the literature. It suggests that the risk of pulmonary fibrosis induced by intra-lesional injection of bleomycin is very low to negligible. Each patient received an injection of 0.5 mg/kg of corticosteroids to prevent this risk.Regarding the use of bleomycin, more details are provided in the response to comment 1. It has been added in the last part of the discussion section.
Furthermore, the manuscript has been double checked by a native speaker for English language.
Round 2
Reviewer 1 Report
1. Response 5. Although you have not seen any volumetric effect of therapy after additional treatment, symptomatic improvement after second therapy cannot be ruled out.
2. Response 6. "As long as the treatment was effective, it was continued until the lesion had completely disappeared. Recurrence was defined as any clinically significant increase in volume."
We know from earlier studies and experience that disappearence of vascular malformation, particularly venous malformation is near impossible.
Similarly recurrence can occur not only by increase in volume but also due to recurrence of symptoms without any change in volume.
3. In Fig.2. it is mentioned as 'Cystic Hemangioma' - It is better to use the terminology 'Macrocystic lymphatic malformation'
Author Response
- Response 5. Although you have not seen any volumetric effect of therapy after additional treatment, symptomatic improvement after second therapy cannot be ruled out.
Response: Thank you very much for your comments. We fully agree. For more understanding, we added the detailed outcomes of second therapy in the 8 patients treated with additional therapy. As previously mentioned, none of them had significant changes in volume, except the patient who received surgical treatment. Among these 8 patients, second therapy led to slight symptomatic improvement in 4 patients based on overall satisfaction evaluation. The 4 remaining patients had no change in their symptoms. It has been clarified at the end of the 3.1 paragraph.
- Response 6. "As long as the treatment was effective, it was continued until the lesion had completely disappeared. Recurrence was defined as any clinically significant increase in volume."
We know from earlier studies and experience that disappearence of vascular malformation, particularly venous malformation is near impossible.
Similarly recurrence can occur not only by increase in volume but also due to recurrence of symptoms without any change in volume.
Response: Thank you very much for your comments. We fully agree. We have the same experience. Our sentence is a language abuse. Besides, none of our patients had complete disappearance of their vascular malformation. The sentence has been rephrased with more nuance, as in our real-life experience, as suggested in the 2.3 section in the 4thparagraph. This new sentence is as follows: “As long as the treatment was effective, it was continued until treatment endpoint as defined above. Recurrence was defined as any clinically significant increase in lesion volume or as recurrence of symptoms whatever the lesion volume”.
- In Fig.2. it is mentioned as 'Cystic Hemangioma' - It is better to use the terminology 'Macrocystic lymphatic malformation'
Response: Thank you very much for your comments. We fully agree. The term “cystic hemangioma” has been changed with “macrocystic lymphatic malformation” in the figure 2, as suggested.
Reviewer 2 Report
I am satisfied with the changes made.
Author Response
I am satisfied with the changes made.
Response: Thank you very much for your comments. No additional changes have been made